# Liquid Biopsy in Colorectal Carcinoma: Clinical Applications and Challenges

**DOI:** 10.3390/cancers12061376

**Published:** 2020-05-27

**Authors:** Drahomír Kolenčík, Stephanie N. Shishido, Pavel Pitule, Jeremy Mason, James Hicks, Peter Kuhn

**Affiliations:** 1Biomedical Centre, Faculty of Medicine in Pilsen, Charles University, 32300 Pilsen, Czech Republic; kolencikdrahomir@gmail.com (D.K.); Pavel.Pitule@lfp.cuni.cz (P.P.); 2Convergent Science Institute in Cancer, Michelson Center for Convergent Bioscience, Dornsife College of Letters, Arts and Sciences, University of Southern California, Los Angeles, CA 90089, USA; sshishid@usc.edu (S.N.S.); masonj@usc.edu (J.M.); jameshic@usc.edu (J.H.); 3USC Institute of Urology, Catherine & Joseph Aresty Department of Urology, Keck School of Medicine, University of Southern California, Los Angeles, CA 90033, USA

**Keywords:** colorectal carcinoma, CRC, liquid biopsy, circulating tumor cell, CTC, circulating tumor DNA, ctDNA, circulating free DNA, cfDNA, precision medicine

## Abstract

Colorectal carcinoma (CRC) is characterized by wide intratumor heterogeneity with general genomic instability and there is a need for improved diagnostic, prognostic, and therapeutic tools. The liquid biopsy provides a noninvasive route of sample collection for analysis of circulating tumor cells (CTCs) and genomic material, including cell-free DNA (cfDNA), as a complementary biopsy to the solid tumor tissue. The solid biopsy is critical for molecular characterization and diagnosis at the time of collection. The liquid biopsy has the advantage of longitudinal molecular characterization of the disease, which is crucial for precision medicine and patient-oriented treatment. In this review, we provide an overview of CRC and the different methodologies for the detection of CTCs and cfDNA, followed by a discussion on the potential clinical utility of the liquid biopsy in CRC patient care, and lastly, current challenges in the field.

## 1. Introduction

Colorectal carcinoma (CRC) is one of the most diagnosed cancers in the world and the second leading cause of cancer related deaths [1]. In high-income countries, or in countries with accessible health care, there are observable stabilizing trends in the incidence and mortality rates of CRC, but overall rates are still one of the highest [2]. Interestingly, adults below 50 years of age are the exception, where the incidence of CRC has increased. In many low-income and middle-income countries, there are distinguishing patterns indicating a rising incidence and mortality rate of CRC [3]. Interestingly, in a projection of global trends in CRC to the year 2035, colon cancer and rectal cancer mortality rates were predicted to decline. However, ongoing demographic changes (population growth and ageing) may lead to a rise in the number of deaths in many countries, with a doubling of the number of predicted deaths by 2035 in some regions [4]. Furthermore, CRC causes a financial strain to a significant number of patients (~40%), which results in a lower quality of life [5]. Overall, CRC can be defined as one of the greatest challenges to public and global health in the present and most likely in the future.

Colorectal carcinoma (CRC) is often diagnosed in late stage due to nonspecific symptoms, such as a change in bowel movement, weight loss, abdominal pain, iron deficiency, anemia, or rectal bleeding [6]. The gold standard for detection of CRC is currently colonoscopy [7]. Furthermore, CRC is clinically categorized by anatomical location as right CRC (RCC) or left CRC (LCC). RCC is defined as the proximal two-thirds of the transverse colon, ascending colon, and caecum [8]. LCC includes the distal third of the transverse colon, splenic flexure, descending colon, sigmoid colon, and rectum. In general, there is a higher incidence of RCC among older patients (Figure 1) [9,10]. Studies comparing screening with and without colonoscopy found a statistically insignificant difference between LCC and RCC [11]. The main concern with RCC is that the right colon has a wider lumen and more frequently flat tumor growths which lead to a longer period without clinical symptoms. Subsequently, this results in a greater time to disease detection and start of treatment [12]. The incidence of stage IV cancer with less differentiated cells is also higher in RCC than LCC [9]. With respect to molecular pathways, the same frequency of the oncogenes Kirsten rat sarcoma viral oncogene homolog (KRAS)and neuroblastoma rat sarcoma viral oncogene homolog (NRAS)are seen in LCC and RCC, but the rate of v-Raf murine sarcoma viral oncogene homolog B (BRAF) mutation has been shown to be significantly higher in RCC [9]. Taken together, RCC is associated with a higher risk of poor prognosis than LCC, despite being classified as the same primary cancer.

Colorectal carcinoma (CRC) is characteristic for wide intratumor heterogeneity and general genomic instability, which impacts the treatment and quality of life of the patient [13]. Accumulation of somatic mutations, which is associated with CRC tumor progression, can be explained with molecular changes that add to genomic instability. Specifically, there are three major molecular pathways in CRC that produce these mutations: chromosomal instability (CIN) [14], microsatellite instability (MSI) [15], and CpG island methylator phenotype (CIMP) [16,17,18]. CIN, as a consequence of improper mitosis and spindle checkpoint activity, promotes tumor progression by increasing the rate of genetic aberrations [19] and it is observed in the majority of sporadic CRC (85%) [20]. MSI is caused by the inactivity of the DNA mismatch repair (MMR) [21] and can be detected in approximately 15% of CRCs [22] and gives the disease distinctive pathological features. Tumors that are positive for MSI tend to be focal, poorly differentiated [23], right-side located, and are associated with production of extracellular mucin [24,25]. CIN and MSI are not mutually exclusive and MSI tumors can show evidence of CIN [26]. MSI is also associated with hereditary non-polyposis colon cancer (HNPCC), the common form of hereditary CRC. The majority of HNPCC-associated mutations affect two crucial genes for MMR, which are MSH2 and MLH1 [27]. Lastly, CIMP positive tumors are defined by transcriptional inactivation by DNA methylation at promoter CpG islands of tumor suppressor genes. CpG islands are regions of the DNA sequence, typically associated with transcriptional promoters, containing a high density of the dinucleotide sequence cytosine followed by guanine in the 5′ to 3′ direction. These tumors have a strong positive correlation to BRAF mutation [18,28]. However, regardless of MSI or BRAF status, CIMP tumors are associated with a significantly lower mortality rate. The relation between CIMP positivity and lower mortality is consistent across all stages of CRC [29]. Specific clones of CpG islands are methylated exclusively in CRC, which has not been observed in normal colon tissue [30]. Undoubtedly, CRC is a well-researched disease from a genomic point of view. Coupled with information about key driver genes, the information creates an ideal background for further research and subsequently utilization of the liquid biopsy in clinical settings.

There are six key driver genes in CRC [31], being APC, TP53, KRAS, BRAF, PIK3CA, and SMAD4, with the TP53 alteration being selectively enriched in the metastatic setting [32]. There are also several signaling cascades which can be affected. The most common is the WNT pathway occurring with most mutated APC genes [33,34]. Overall, WNT pathway alteration occurs in 93% of MSI positive and 85% of MSI negative CRC [32]. The combination of mutations in the WNT pathway and in the signaling pathway associated with KRAS mutation are crucial for tumor progression in CRC [35,36]. A KRAS mutation (and BRAF mutation) drives tumorigenesis through constitutive activation of the MAPK pathway [37] and can induce hyperproliferation in colonic epithelium, but only in combination with a mutation from the WNT pathway [38]. KRAS mutations are also associated with the PI3K pathway [39], which plays a fundamental role in the tumor–host interaction, enhancing tumor-induced angiogenesis and facilitating the establishment of metastatic colonies [40]. BRAF mutations are present in 5–13% [41,42,43] of sporadic CRC (geographical variation may account for differences in the occurrence [43]) and are significantly associated with a higher metastatic rate and worse overall survival (OS) [44]. Subsequently, BRAF mutation can be used as a prognostic factor and in clinical decision-making regarding targeted therapy [41,42]. Additionally, since BRAF is a downstream molecule of KRAS [43], concomitant KRAS and BRAF mutations are rare and could be considered mutually exclusive [45]. Correspondingly, combined mutation of PIK3CA and TP53 is correlated with a shorter OS of stage II/III CRC patients receiving 5-fluorouracil-based therapy [46].

Colorectal carcinoma (CRC) is a malignant disease with severe impact on the general population. Genomic information of CRC implies the challenge of intratumor heterogeneity and related poor outcomes during treatment. Extensive information presented in this review will demonstrate the promising opportunity for the liquid biopsy to improve prognosis and patients’ quality of life.

## 2. Liquid Biopsy for CRC

The liquid biopsy is generally referred to as the analysis of tumor-derived material from peripheral blood. However, in the general sense, this can also include sampling from other bodily fluids such as urine [47], stool [48], cerebrospinal fluid [49], saliva [50], pleural fluid [51], and ascites [52]. In this review, circulating tumor cells (CTCs) and cell-free DNA (cfDNA) derived from peripheral blood will be the focus of this discussion.

### 2.1. Circulating Tumor Cells

Circulating tumor cells (CTCs) may be detectable in the cellular fraction of patient peripheral blood. CTCs are defined as tumor cells in circulation coming from the primary tumor or from a metastatic site [53]. On average, CTCs measure between 15 and 25 µm, which is generally larger [54,55] than white blood cells (6–10 µm) [56], and have a distinct morphological shape of the nucleus [57]. Although CTCs were discovered 150 years ago [58], it was not until the development of novel detection methods that they became the focus of new prognostic and diagnostic approaches. Individual CTCs can be characterized by their morphology [59], phenotype, and genotype [60]. Genomic information from CTCs may be used to better understand cancer cell biology, provide confidence around diagnostic [61] or treatment decisions [62], and to track treatment response [63]. Genomic information also allows for the assessment of clonality and the characterization of different cell populations [64,65]. CTCs can be detected in the peripheral blood as single cells or as cell clusters [66]. Steeg hypothesized that tumor cells invade the tissue surrounding the primary tumor, enter the lymphatics or the bloodstream, circulate in the body, then extravasate into a tissue, and form a secondary tumor at the new location [67]. During this metastatic process, CTCs can interact with immune cells [68]. For example, CTCs that express programmed death-ligand 1 (PD-L1) [69] may interact with circulating immune cells and have been associated with an increased two-year mortality risk [70].

### 2.2. Cell-Free DNA

The discovery of free DNA in blood dates back to 1948 by Mandel and Metais [71]. Typically, cfDNA in healthy individuals derives from the normal turnover of cells through apoptosis [72] or necrosis [72], or active release from lymphocytes [73,74,75] and can be increased by such events as infection or inflammation [76]. In cancer patients, the level of cfDNA can be increased through the turnover of tumor cells. This component can be distinguished from normal cfDNA by the presence of tumor-associated mutations and is often categorized separately as circulating tumor DNA or ctDNA. Although cfDNA can be found as fragments from 150 to 10,000 bp, the vast majority is found at a peak of 166 bp, corresponding to the size a single nucleosome, as is typically released during apoptosis [77,78]. Peripheral blood of a healthy individual contains only limited amount of cfDNA (up to 100 ng/mL) [79]. In contrast, the patients with metastatic CRC have been shown to have significantly higher levels of cfDNA in blood (up to 209 ng/mL) [79]. Recently, the same relationship was shown again in which patients with primary CRC had significantly higher levels of cfDNA than patients with intestinal polyps and healthy controls [80]. Analysis of cfDNA in CRC can be used for testing KRAS and BRAF mutations. In the case of KRAS mutation, cfDNA analysis may identify patients with a worse prognosis who may benefit from a more aggressive chemotherapy regimen [81]. Additionally, analysis of the BRAF mutation can be used as a negative prognostic biomarker for OS [82]. Likewise, the detection of BRAF mutation in cfDNA may have a role in treatment selection for patients [83].

## 3. Liquid Biopsy Platforms

### 3.1. Detection of Circulating Tumor Cells

The platforms used for detection and collection of CTCs vary widely in their technology and methodology. They can be separated into four different categories based on their technique of enrichment: immunocapture, physical characteristics (i.e., size and density), non-enrichment based, or a combination. To embrace all of the possibilities of how CTCs can be utilized in clinical routine, it is necessary to understand the advantages and potential biases of those different technologies. Immunocapturing is one of the most common used techniques of detection and is based on targeting specific antigen(s) on the surface of cells. A positive enrichment immunocapturing approach, often based on the epithelial cell adhesion molecule (EpCAM) antigen, targets the cells of interest. On the other hand, negative enrichment, typically based on the CD45 antigen, targets the removal of other cell populations, typically lymphocytes, and thereby enriches the cells of interest. In either case, only cells displaying these antigens are detected and consequently either collected or discarded. Techniques based on physical properties of the cells are utilizing the size and or density difference between CTCs and other non-rare cells. As with immunocapturing, only cells with certain physical properties are collected. For this reason, both techniques carry inseparable bias. Non-enrichment techniques do not exclude cells from a sample. This approach presents the possibility of avoiding potential bias during the identification process linked to the physical or biological properties of cells. Consequently, non-enrichment techniques can be more labor-intensive than the other methods. Further discussion about possible limitations of the liquid biopsy can be found in Section 5, Challenges.

A list of some current platforms is provided in Table 1. However, the field of liquid biopsy is rapidly developing and therefore it is not our intention to provide an exhaustive list of all platforms. In addition, brief details of three different approaches to detect CTCs can be found below. The specific platforms which are discussed were selected as an example of each enrichment technique to illustrate their practical use.

#### 3.1.1. CellSearch^®^

The CellSearch^®^ platform (https://www.cellsearchctc.com/) is based on immunomagnetic enrichment of cells expressing EpCAM from peripheral blood samples. The circulating epithelial cells are selected for using immunomagnetic beads targeting EpCAM. The enriched sample is then stained with 4′,6-diamidino-2-phenylindole (DAPI) and immunofluorescently labeled with a monoclonal antibody for CD45 to identify leukocytes, and with antibodies against cytokeratins (CK) 8, 18, 19 to identify epithelial cells. CTCs are then defined as CD45 negative and CK positive [84,85,86]. CellSearch^®^ was the first approved CTC detection and capture platform by the U.S. Food and Drug Administration (FDA) for prognostics in cancer treatment and has been allowed for clinical use in breast [87], colorectal [85,88], and prostate cancer [86,89]. A prospective study using CellSearch^®^ analyzed the dynamic change of CTCs between pre- and post-surgery 7.5 mL blood samples collected in a cohort of 44 metastatic CRC patients (43 pre-surgery and 38 post-surgery samples). Results showed that all patients with pre-surgery positive samples (two or more CTCs per sample) relapsed. In contrast, 65% of patients with pre-surgery negative samples relapsed. Similarly, 68% and 85% of patients with post-surgery negative and positive samples, respectively relapsed. The study also showed significant difference in median OS between patients with positive and negative pre-surgery samples with 17 and 69 months, respectively [90].

#### 3.1.2. Epic Sciences/High-Definition Single-Cell Assay

The high-definition single-cell assay (HD-SCA) workflow is a non-enrichment method developed for high-resolution characterization of CTCs, which provides intact cells enabling downstream single cell molecular characterization [57,91,92,93]. Biospecimens are processed within 24 to 48 h after collection, which involves erythrocyte lysis and centrifugation. The HD-SCA workflow is a “no cell left behind”™ approach in which all nucleated cells from the blood sample are plated on custom glass slides, approximately 3 million nucleated cells per slide. Slides are subsequently stained with DAPI to identify the nuclei and antibodies against pan CK and CD45. There is the possibility of adding a 4th channel for further characterization of the cellular populations. This approach allows for the functional profiling of all nucleated cells from a patient’s blood without enrichment bias [92,94] based on their morphological features, biomarker expression, and nuclear integrity [95]. A study using this platform, with a primary goal of establishing concordance among morphology of liquid and solid biopsy cells, gave substantial evidence by analyzing more than 1000 single cells. Out of 43 metastatic CRC patients, 15 patients had more than 4 HD-CTC per mL (CK positive, CD45 negative, morphologically distinct cells with a nuclear size larger than surrounding white blood cells) (35% positivity) in their pre-surgery blood. Additionally, higher concentration of HD-CTC was observed by patients with necrotic hepatic tumors. The level of HD-CTCs also decreases in post-surgery blood of patients. The results demonstrated that liquid biopsy cells are associated with solid tumor cells in metastatic CRC and the relationship can be potentially used for diagnostics or monitoring disease burden over time [92].

#### 3.1.3. The Isolation by Size of Epithelial Tumor Cells^®^

The isolation by size of epithelial tumor cells (ISET^®^, https://www.isetbyrarecells.com/iset-story/) platform is based on size enrichment, which operates on the presumption that tumor cells derived from carcinomas have a significantly larger size compared to peripheral blood leukocytes [96]. Blood samples are filtered through a module with up to 12 wells, where each well contains a 0.6 cm diameter membrane [96] with 8 μm diameter cylindrical pores [97]. Each well can then be further analyzed with immunohistochemical characterization of cells isolated by ISET^®^ [59], fluorescence in situ hybridization, or genomic sequencing [98]. A study was conducted to establish correlation between ISET^®^ CTC count and both progression-free survival (PFS) and OS in patients with metastatic CRC. Three serial collections were analyzed via ISET^®^ and the CTC number varied from 0 to 46 cells per sample. Based on the variation in number of CTCs found between the collections, it suggests possible risk stratification [99].

### 3.2. cfDNA Analysis

Measurement of cfDNA, and subsequently ctDNA (circulating tumor DNA) detection, can be challenging due to low concentration, especially in early stages of cancer [128]. Generally, ctDNA can be analyzed with a focus on tumor-specific mutations [129], genome wide analysis for copy number alterations (CNAs) [130,131], point mutations by whole-genome sequencing (WGS), or whole exome sequencing (WES) [132]. In addition, CNAs are theoretically easier to detect than point mutations or epigenetic changes in cfDNA [133], because CNA analysis requires only sparse sequence coverage [131]. cfDNA-based CNAs show clinical validity as promising biomarkers for cancer diagnosis and prognosis, especially for late-stage cancers [134].

An important aspect in cfDNA targeted mutational analysis is mutant allele fraction (MAF). MAF is defined as a ratio between mutant alleles and all targeted alleles in a sample. Thus, sensitivity of MAF is a significant characteristic for platforms detecting ctDNA. In a case of cfDNA analysis, targeted methods are more suitable and applicable in the clinical setting. Classical quantitative PCR (qPCR) methods have sensitivity of between 10–20%. Advanced targeted methods for detecting MAF in cfDNA are digital PCR (dPCR), which include droplet digital PCR (ddPCR) [135,136] and BEAMing (beads, emulsions, amplification, and magnetics) [137]. Studies that used BEAMing were able to detect MAF of 0.1% [138,139]. Detection sensitivity of ddPCR assays is 0.04% [140]. Furthermore, ddPCR is not solely used for point mutation detection, but is able to reveal indels and frequently observed cancer-related CNA [141]. Through PCR enrichment, with suitably short amplicons, amplicon-based sequencing can achieve sensitivity comparable to ddPCR [142]. Tagged-amplicon deep sequencing (TAm-Seq) technology and its enhanced version could detect MAF of 0.14% and 0.02%, respectively [142,143]. Another targeted sequencing method able to detect cfDNA in small amounts is duplex sequencing with unique molecular identifiers, which can distinguish MAF of 0.1%. The duplex method exploits the knowledge of complementarity of both DNA strands. Adapters tag duplex DNA, and afterwards, it is possible to compare the strand with its counterpart. Mutation originating from the tumor must be present in both strands [144]. Another targeted sequencing method is cancer personalized profiling by deep sequencing (CAPP-Seq) [145] or safe-sequencing system (Safe-SeqS) [146]. CAPP-Seq combines identifying recurrent mutated regions in given cancer type via bioinformatic methods and sequencing with the objective to improve sensitivity and approximate patient-orientated approach [147]. CAPP and its enhanced methods are able to detect MAF of 0.2% and 0.04%, respectively [148]. Safe-SeqS is similar to duplex sequencing by assigning an unique molecular identifier. In contrast to duplex sequencing, these identifiers tag each template molecule. Safe-SeqS is able to detect MAF of 0.01% [149].

On the other hand, non-targeted methods for cfDNA analysis may harbor some advantages. They do not depend upon specific knowledge about the primary tumor, neither genetic nor epigenetic. However, genome-wide sequencing methods, like whole-genome sequencing or whole-exome sequencing, require a relatively high fraction of ctDNA/cfDNA for detection (5–10% MAF) [150]. For this reason, these methods are better suited for research than clinical routine.

There are some commercial cfDNA assays which can be used for clinical applications in CRC. In 2016, the FDA granted premarket approval for the first cfDNA-based liquid biopsy for cancer screening, Epigenomics’ Epi proColon^®^ assay, using qPCR. This assay is based on the presence of aberrantly methylated SEPT9 DNA in the plasma and has shown promising results [151]. Out of 50 untreated CRC patients prior to surgery, the assay was able to detect positive 45 patients (90% sensitivity) for presence of methylated SEPT9 DNA. More interestingly, the assay was able to detect 33 out of 38 (87% sensitivity) patients with early stage disease (stages I and II) [152]. Since then, another cfDNA assay, Signatera™, received breakthrough device designation by the FDA.

Another commercial platform focused on CRC is the Idylla™, also using qPCR assays for KRAS, [153], NRAS, and BRAF mutations [153] or characterization of MSI [154]. The platform OncoBeam™, using BEAMing (beads, emulsification, amplification, and magnetics), offers panels for the detection of various CRC specific mutations, including KRAS, NRAS, BRAF, and PIK3CA [155]. Another interesting approach was taken by Wan et al. from Freenome, which relied on a machine learning approach using cfDNA from 546 CRC patients, the majority of which were in stage I and II (81%), to develop a computational approach to identify relation between the cfDNA profile and CRC stage. Blood samples were collected at unspecified timepoints. A mean sensitivity of 85% was achieved, showing promising preliminary performance for detection of CRC in early stages. [156].

## 4. Clinical Applications

Liquid biopsies can be utilized in patient-orientated treatment and precision oncology, which can be especially important when spatial and temporal intratumor heterogeneity is connected to the rise of distinct resistance mechanisms in different metastases. That may subsequently lead to lesion-specific responses of targeted therapy [157]. Furthermore, minimal residual disease (MRD) represents potential risk for every patient in curative treatment since the presence of cancer cannot be found by standard clinical means. The assessment of MRD is vital for adequate early treatment planning [158]. Liquid biopsies have the potential to predict disease recurrence even when clinically undetectable. In clinical practice, there are already a few methods which utilize similar approaches to the liquid biopsy. Protein-based circulating biomarkers, such as carcinoembryonic antigen (CEA), are currently clinically measured and used to assist in diagnostic [159] and prognostic decisions [160]. High postoperative CEA levels increase the chance of relapse and decrease OS [161]. Correspondingly, the ratio of CEA pre and post-surgery levels has been found to be a reliable prognostic factor in stage IV CRC [162]. As mentioned briefly above, stool samples from CRC patients can be used for further analysis. KRAS mutations have been detected in DNA purified from the stool samples [48]. The platform Cologuard^®^ (https://www.cologuardtest.com/meet-cologuard) received premarket approval by FDA in 2014 and was developed for screening early CRC, targeting mainly β-actin, KRAS mutations, and aberrantly methylated BMP3 and NDRG4 [163].

This shows that the liquid biopsy is already in use and current research is aiming to increase their clinical utility. Here, we will discuss the potential clinical applications and challenges for either CTCs or cfDNA in the context of CRC.

### 4.1. Diagnostics

An accurate diagnosis is crucial for making clinical decisions regarding the best course of treatment. The liquid biopsy can serve as a first step in the screening program for CRC, triggering follow-up examinations aimed at the early detection of invasive cancers [164,165]. The importance of the liquid biopsy may provide critical clinical insight into molecular subtypes of the tumor, especially when the discordance in KRAS mutations between primary and recurrent tumors after resection can be as high as 20% [166]. The intratumor heterogeneity and its relevance in CRC was further discussed in additional studies. Fabbri et al. first demonstrated the feasibility of analyzing pure CTCs at the molecular level and avoiding lymphocyte contamination using the DEPArray, a dielectrophoresis-based platform, as well as a KRAS discordance between CTCs and primary tissue cancer after 100% pure cell recovery and sequencing. In a cohort of 40 metastatic CRC patients, there were 21 patients with more than three CTCs in a sample of 7.5 mL of blood. Additional KRAS analysis of 16 patients showed only 50% concordance between primary tumor tissue and CTCs [167]. Another study of Russo et al. in CRC cell culture discussed the possibility of induced heterogeneity by targeted therapy. Results showed that cells, which survive targeted therapy based on EGFR and/or BRAF inhibition, tend to show more DNA damage, down-regulate DNA proteins for mismatch repair, homologous recombination protein, and temporarily increase the chance of mutation [168]. Additionally, in another study, molecular events connected to resistance to certain targeted treatment were described, especially regarding KRAS and BRAF [169].

Analysis of cfDNA offers the potential to provide molecular information and confidence around early diagnosis. In a recent study, there was an investigation of a novel potential use of ctDNA methylation markers. Plasma samples were collected from 801 CRC patients of various stages. Consequently, they analyzed 544 methylation markers, created diagnostic scoring and compared it to the pathological diagnosis. Sensitivity and specificity of the diagnostic scoring was 87.5% and 89.9%, respectively [170]. Despite this promising correlation between pathological diagnosis and cfDNA, cfDNA in CRC must prove clinical utility before it can be used for early screening or detection [171,172].

The clinical significance of CTCs has not been as extensively explored as cfDNA. There are few studies which analyze agreement of CTC enumeration and morphology with TNM staging (staging based on tumor size (T), spread of cancer to nearby lymph nodes (N) and metastasis (M)). Eliasova et al. using the MetaCell^®^ platform (http://metacell.cz/about/) utilizing a size-based separation method [173], showed high concordance of the TNM stage with the number of CTCs in each patient sample [174]. In this cohort of 98 CRC patients, 88.89% of patients with colon cancer had CTCs in their blood samples while 77.36% patients with rectosigmoid cancer had detectable CTCs. This study also showed a positive correlation between the number of CTCs in patients’ blood and the size of the primary tumor [174]. In another study using the CanPatrol™ platform (SurExam, Guangzhou, China), the presence of CTCs in the peripheral blood was correlated with disease stage. Blood samples of 1203 patients were collected at baseline, i.e., prior to surgery in the case of curative operation, before treatment in the case of palliative resection, or during chemotherapy treatment intervals in the case of advanced stages. CTCs were divided into three groups based on epithelial (EpCAM, CK 8, 18, and 19) and mesenchymal markers (VIM, TWIST1, AKT2, SNAI1). Both detection rate and the mean number of mesenchymal cells or cells with combined epithelial and mesenchymal markers showed concordance with stages of CRC. As an example, 481 patients had one or more CTCs of the epithelial phenotype (per 5 mL blood), 684 patients had one or more CTCs of the mesenchymal phenotype, and finally, 924 patients had one or more CTCs displaying both epithelial and mesenchymal phenotypes. Interestingly, most segregated distribution and positive correlation regarding metastatic status was demonstrated by CTCs of the mesenchymal phenotype [103]. Another interesting approach was taken to analyze morphological concordance of liquid to solid biopsy. A study using the HD-SCA workflow displayed the relationship between the liquid and solid biopsy by cellular morphology in metastatic CRC patients. In a cohort of 43 metastatic CRC patients, a total of 1058 CRC cells from either pre-surgery blood sample or solid biopsy of hepatic metastasectomy were examined. The results displayed an overall similar intra-patient morphology assessed using hierarchical clustering of certain features (i.e., object size, shape, or biomarker distribution) normalized within each patient [92].

### 4.2. Treatment Selection

The liquid biopsy can have an important practical impact for the treatment of patients. Precise and ongoing molecular characterization of CRC is crucial for correct use of molecular targeted therapies. KRAS and NRAS mutations in general vary widely between sporadic CRC lesions, and the status for those mutations in metastases is unpredictable [175]. A liquid biopsy can be used to detect KRAS mutations in cfDNA in the absence of detection from a primary tumor biopsy. This can be important in treatment selection since KRAS mutated cancer cells are resistant to treatment with anti-EGFR monoclonal antibodies treatment [176]. A study using the OncoBeam™ (https://www.oncobeam.com/healthcare-providers/colorectal-cancer) RAS CRC assay showed that the overall agreement between solid and liquid biopsy was 96.4%. Out of 55 patients positive for the RAS mutation in the tumor tissue, 53 patients had the RAS mutation in cfDNA as well [177]. With the same assay, an additional study, with a cohort of 236 metastatic CRC patients, shows an 89% correlation of the RAS mutation between the solid biopsy and cfDNA [155]. Another study assessing the clinical utility of cfDNA, with a cohort of 140 metastatic CRC patients, showed slightly different results. Only moderate concordance (72–87% accuracy) between plasma samples and tumor tissue was observed, potentially due to the higher frequency of the KRAS mutation in the plasma samples [178].

A prospective phase II study found that approximately 50% of the metastatic CRC patients on trial that were identified without KRAS mutations after the tumor biopsy were shown to actually contain RAS aberrations and did not benefit from anti-EGFR monoclonal antibody treatment [179]. Additionally, Klein-Scory et al. observed the emergence of RAS mutated subclones during therapy with the anti-EGFR monoclonal antibodies in three mCRC patients who were initially determined to be RAS wild-type [180]. In a separate study, overall concordance was 78.8% between the KRAS genotype in the primary tumor and cfDNA sample in a cohort of 52 patients with histologically confirmed CRC and undergoing surgical treatment resection [181]. In a similar study by Bettegowda et al. [182], the sensitivity of ctDNA for testing KRAS was 87.2% in a cohort of 206 patients with metastatic CRC, where in addition to 69 patients with circulating KRAS mutation, 10 cases had detectable mutation in the primary tumor but not in the plasma. This information helped to understand the otherwise inexplicable outcome when patients without RAS mutations detected in the tissue biopsy do not respond to targeted therapy [182].

Temporal and spatial intratumor heterogeneity of CRC represents a great challenge for current treatment decision-making and it can only currently be conveyed through an invasive solid biopsy. The liquid biopsy offers a feasible, non-invasive, and patient-orientated approach to addressing intratumor heterogeneity and may provide valuable information that could increase efficacy of targeted therapies and quality of life of patients.

### 4.3. Prognostics

Another potential clinical use of the liquid biopsy is for prognostication. Its utility has been shown in various studies and meta-analyses [183,184,185,186]. Molecular profiling of the liquid biopsy may produce critical information to provide confidence around assessing patient prognosis. The level of cfDNA is significantly higher in patients with a poor treatment response and a significantly shorter PFS and OS compared to patients with low levels of cfDNA [187]. Patients with KRAS mutations detected in the plasma correlate with shorter PFS and OS. Combining the information around total cfDNA load and KRAS mutation status from the liquid biopsy provides additional prognostic effect [187]. The prognostic potential of cfDNA, specifically ctDNA, was further demonstrated in another study with cohort of 130 CRC patients in various stages. Out of 456 postoperative samples of patients without relapse, 455 samples were ctDNA negative. In comparison, 14 out of 16 patients with disease relapse had detectable ctDNA. Furthermore, 30% of patients who responded to chemotherapy, had no detectable cDNA and remained cDNA negative, stayed disease free [188].

As mentioned above, the first FDA approved CTC assay for colon cancer was CellSearch^®^ (https://www.cellsearchctc.com/). This assay is able to stratify metastatic CRC patients into favorable (less than three CTCs/sample) and unfavorable (three or more CTCs/sample) prognostic groups based on CTC levels [189,190]. A recent study from 2019 showed that CellSearch^®^ detected at least one CTC per 7.5 mL in 25 patients from a cohort of 80 CRC patients in various stages (31.3% positivity). In all patients, blood was collected before surgery together with routine clinical samples. Subsequent survival analysis showed that the presence of three or more CTCs was significantly associated with poor OS. Analogously, in the subgroup of 29 patients with nonmetastatic disease, detection of one or more CTCs correlated significantly with a worse OS in the univariate analysis [191]. In contrast, it has been shown that in a cohort of 344 patients with non-malignant diseases or healthy subjects, there was only one sample with more than one CTC [84]. Despite the proven prognostic value, it is rarely used in routine clinical care [92]. Furthermore, a meta-analysis based on 15 studies (including detection methods such as CellSearch^®^, RT-PCR, flow cytometry, membrane array, and quantitative immunofluorescence) confirmed prognostic value in the presence of CTCs, correlating with worse PFS and OS [186]. However, additional studies into the clinical utility of CTCs in CRC prognostication are still warranted.

## 5. Challenges

Utilizing ctDNA and CTCs can offer new methods of diagnosis, prognosis, following treatment response, and most importantly, the liquid biopsy platforms aim to provide necessary information to improve patients’ outcome. Nevertheless, challenges like pre-analytical variables, rarity of CTCs and ctDNA in samples, analytical validity, clinical validation, cost effectiveness, and regulatory approval need to be addressed prior to clinical utilization. With respect to pre-analytical variables, using liquid biopsy in day-to-day routine requires verified sample collection protocol—how to collect, store, transfer and process samples [192]. cfDNA can be sensitive to improper handling, thus compromising the quality of extractable DNA and integrity of the downstream mutational analyses [181]. CTC detection is significantly impacted by pre-analytical variables, such as type of blood collection tube, time-to-assay [93], and the optimal sampling time of peripheral blood [186]. Recently, efforts have been made to standardize pre-analytical workflows for liquid biopsy in context of the European Union’s Horizon 2020 SPIDIA4P consortium project, which suggests existing demand for a validated workflow [193]. Regarding rarity, liquid biopsy is problematic from the technological point of view. For example, the small size and distribution of ctDNA fragments makes it difficult to detect certain important chromosomal aberrations such as gene fusions [194]. In the case of CTCs, very low numbers of these cells in the peripheral blood makes detection similarly challenging [92,195]. The relative rarity of these elements needs to be to be taken into account during development of detection platforms and assays and consequently during their clinical validation. Pre-analytic limitation could be also represented by volume of the sample. In the study of Dizdar et al., the CellCollector^®^ device was evaluated in comparison to CellSearch^®^. This device aimed to bypass the limitation of the low sample volume by inserting a wire covered with EpCAM antibodies through a placed 20-G venous catheter into the cubital vein of the patients to capture CTCs directly from the peripheral blood. In contrast, CellSearch^®^ requires collection of 7.5 mL blood for analysis. Nevertheless, the rate of CTC detection was similar by both platforms with at least one CTC detected by CellCollector^®^ (https://gilupi.com/our-products/?lang=en) in 33 patients from a cohort of 80 CRC patients (41.3% positivity) [191].

Many methods which are based on the positive capture of CTCs, especially those relying on EpCAM, are intrinsically biased and potentially underestimate the true number of CTCs. These methods can also miss potentially important relationships between and within cells that do not express EpCAM [196]. Additionally, the efficiency of positive enrichment methods that rely on certain biological properties, such as surface marker expression or size, can be reduced by dynamic cellular changes, such as epithelial-to-mesenchymal transition [197].

A main concern for analytic validity of the liquid biopsy is accuracy and reproducibility. One recent study raised the question of the reliability of commercially available single-nucleotide variant (SNV) panels for ctDNA analysis. Two panels, PlasmaSELECT™ and Guardant360™, were examined to determine the reliability and potential utility of ctDNA in the clinical world. The study showed very low congruence for the same patient-paired samples in these two commercially available tests [198]. Similarly, the results of another study demonstrated different levels of detection of the KRAS mutation in the same plasma samples between two platforms, OncoBEAM™ RAS CRC and Idylla™ ctKRAS Mutation Test. This shows that not all platforms of the liquid biopsy are equivalent, and highlights the need for standardization [199]. Consequently, the liquid biopsy holds great promise, but still faces issues with standardization and validation [193].

Rapidly developing new technical solutions and standardization efforts are ongoing with the goal of implementing a meaningful liquid biopsy analysis into routine use for CRC patient care. Prior to market adoption and clinical implementation, a platform needs to provide proof of clinical utility by showing that its application can improve patient outcomes. For that, analysis needs to be validated, analytically and clinically. Analytic validation lies mainly in reproducibility and consistency of the analysis. For clinical validation, a platform must be optimized for a specific context of use (i.e., cancer type, stage, treatment), with data driven clinical utility for a proposed analyte. Besides cost of the liquid biopsy, clinical utility is the most significant deciding factor which affects reimbursement, and consequently, actual application [200]. Moreover, specific analytes such as CTCs and cfDNA will most likely require separate validation for clinical utility in different disease settings. The challenge is identifying a meaningful context of use for the specific analyte to be applied to predict patient outcomes. To do this requires full analytical validation that the assay is fit for the purpose indicated. This is followed by clinical validation to generate the evidence needed to prove the assay output is associated with clinical outcomes [95,201,202].

Furthermore, cost effectiveness is a major consideration. Since one of the theoretical advantages of the liquid biopsy is the ability to follow temporal intratumor heterogeneity and evolution, it raises the question of the number of necessary follow-up analyses and the related costs [203]. Currently, most technologies are labor-intensive, from library construction to bioinformatics evaluation [204,205]. In an ideal scenario, CTCs and ctDNA would be analyzed in one comprehensible cost-effective way to ensure accurate, reproducible, and truly meaningful results to impact the clinical world. There is also a need for a comprehensive liquid biopsy strategy that would run regularly in larger patient cohorts. Utilization of the complete liquid biopsy would allow us to exploit CTC and cfDNA information at the same time. Functional profiling of CTCs provides an overview of morphological features, biomarker expression, nuclear integrity, and genomic heterogeneity. On the other hand, cfDNA offers valuable data on the bulk genomic profile and point at the most common genomic profile. The combination of both compartments would provide unique impactful results and further develop a patient-oriented approach and precision medicine.

## 6. Conclusions

Incidence, mortality, age of diagnosis, nonspecific symptoms, and intratumor heterogeneity in CRC demonstrates that there is still room for improvement in clinical management and patient outcomes. The liquid biopsy could be the tool that will add a new perspective for clinal routine and confidence around clinical decision-making. The standard tissue biopsy is critical for the pathological evaluation of the tumor at the time of solid biopsy collection and displays current pathological status of the specific lesion. The liquid biopsy is ideal for longitudinal monitoring of advanced disease by molecular characterization, with the additional opportunity of understand the spatial and temporal heterogeneity of CRC [206]. The liquid biopsy could improve diagnostics, prognostics, and treatment response by providing valuable information about the patient-specific disease to assist in clinical decision-making.

## Figures and Tables

**Figure 1 cancers-12-01376-f001:**
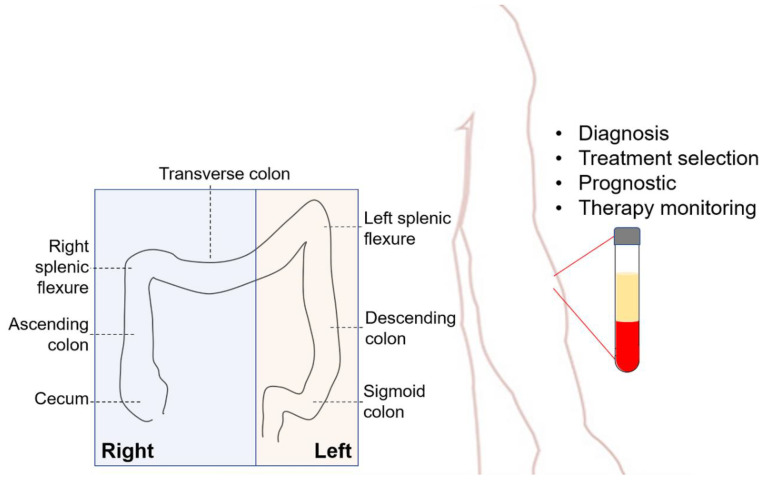
Colorectal carcinoma (CRC) is categorized based on its anatomic location. Right CRC (RCC) is localized in caecum, ascending colon or two proximal thirds of transverse colon. Left CRC (LCC) is defined as CRC in distal third of transverse colon, descending colon, sigmoid colon, or rectum. Clinical applications of the liquid biopsy in CRC include diagnosis, treatment selection, prognostic, and therapy monitoring.

**Table 1 cancers-12-01376-t001:** Platforms used for the detection of circulating tumor cells (CTCs) in the peripheral blood. Three discussed platforms in text are marked with * EpCAM, epithelial cellular adhesion molecule; CD45, cluster of differentiation.

Platform	Company/Institution Details	Description	References
AdnaTest	Qiagen GmbH, Hilden, Germany	Antibody targeting EpCAM conjugated to magnetic beads for labeling tumor cells in sample	[100,101]
CanPatrol™ CTC	SurExam, Guangzhou, China	Filtration (and CD45+ depletion) ^1^	[102,103]
CellCollector^®^	Gilupi GmbH, Postdam, Germany	Nano guidewire inserted into patient cubital vein collecting cell expressing EpCAM	[104,105]
CellMax CMx	CellMax Life Inc., Sunnyvale, CA, USA	Blood passing through antibody-coated microfluidic chip targeting EpCAM	[106,107]
CellSearch^®^ *	Menarini Silicon Biosystems Spa, Castel Maggiore, Italy	Antibody targeting EpCAM conjugated to magnetic beads for labeling tumor cells in sample	[108,109]
ClearCell^®^ FX	Genomax Technologies, Bangkok, Thailand	Blood passing through microfluidic biochip with larger cells along the inner wall	[110,111]
Cytelligen^®^	Cytelligen Inc., San Diego, CA, USA	Antibody targeting CD45 conjugated to magnetic beads for labeling tumor cells in the blood sample	[112,113]
DEPArray™	Menarini Silicon Biosystems Spa, Castel Maggiore, Italy	Cell suspension loaded into microchip-based sorter using dielectrophoresis to trap cells	[114,115]
Easysep™	Stemcell Technologies Inc., Vancouver, BC, Canada	Antibody targeting EpCAM or CD45 conjugated to magnetic beads for labeling tumor cells in the blood sample	[116,117]
Epic Sciences/HDSCA *	Epic Sciences Inc., San Diego, CA, USA	After processing, cells are plated on the slide and subsequently characterized based on surface markers	[57,92]
Herringbone Chip	Massachusetts General Hospital, Boston, MA, USA	Blood processed through antibody-coated microfluidic chip targeting EpCAM	[118,119]
ISET^®^ *	Rarecells Diagnostics SAS, Paris, France	Filtration on pressure-controlled system	[120,121]
MagSweeper™	Stanford University, Stanford, CA, USA	Antibody targeting EpCAM or CD133 conjugated to magnetic beads for labeling tumor cells in blood	[122,123]
MetaCell^®^	MetaCell s.r.o., Ostrava, Czech Republic	Capillary-action driven size-based separation	[124,125]
Oncoquick^®^	Greiner Bio-One International GmbH, Kremsmünster, Austria	The denser blood compartment migrates through the porous barrier of the polypropylene centrifugation tube	[126,127]

^1^ Negative immunocapturing step was left out in some studies.

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
