# Peer review of "Liquid Biopsy in Colorectal Carcinoma: Clinical Applications and Challenges"

_cancers, 2020, doi:10.3390/cancers12061376_

Round 1

Reviewer 1 Report

Well written review of the clinically applicable technology for circulating tumor cells and cell free DNA in colorectal cancer. Comprehensive and critical review of the data available in the field. 

As a suggestions, the authors may consider including data on Cologuard in this review, as a clinical test of major importance and utility in CRC. 

Author Response

We would like to express our gratitude to all the reviewers for their detailed reports. We found your comments and suggestions valuable and overall, they helped to increase the factual and stylistic precision of the text. Particularly, we have focused on improving the general understanding and logical coherence of the underlying theme of the review. Please see below for our direct replies.

Reviewer #3

Well written review of the clinically applicable technology for circulating tumor cells and cell free DNA in colorectal cancer. Comprehensive and critical review of the data available in the field. 

As a suggestions, the authors may consider including data on Cologuard in this review, as a clinical test of major importance and utility in CRC.

RESPONSE: Thank you for your suggestion. We have added Cologuard as an example of cfDNA assay used in CRC in section 4. Clinical applications.

Reviewer 2 Report

General comment

The review of Kolenčík and colleagues aimed at revising the literature on CTCs in colon cancer. In the liquid biopsy research area, colon cancer is effectively a less frequented topic; hence, the review could be an opportunity to do the point on state of art of liquid biopsy in this malignancy. However, in the opinion of this reviewer, to address this purpose some gaps need to be filled before publication.

Speaking about the different technologies, the authors lack to summarize the main strategies of liquid biopsy procedures, and their pros and cons. Moreover, sometime ignore some fundamental findings.

For example, the authors choose to present a Table with the list of the technologies for CTC detection; furthermore, they discuss more in depth only three techniques. It might be a good way, to give a simple and precise picture of the state of art. Unfortunately, the reader is not warned that the list is largely incomplete (e.g. authors forget to cite DepArray), since liquid biopsy is a hot topic, and monthly several new techniques are published. Moreover, the authors do not explain why they choose to discuss CellSearch, ISET and Epic Science technologies. Overall, the section about CTC detection techniques results poor. 

Furthermore, despite the authors underline the value of liquid biopsy to investigate temporal and spatial tumor heterogeneity, they lack to discuss some sound papers, e.g. those of Fabbri and Bardelli, in colon cancer.

Second, in the opinion of this reviewer the clinical section of the review is not well organized; moreover, there are semantic imprecisions in description of clinical activity, hence inducing misleading conclusions in the reader. For example, the first paragraph is entitled “Diagnostics”; in this paragraph, there is an overlap between early diagnosis and early prediction of treatment effectiveness, that are different uses of biomarker. By the way, there are no prospective studies about the diagnostic use of liquid biopsy in colon cancer, whilst the use of liquid biopsy as predictor of treatment effectiveness has been partially entered in colon cancer management.

Finally, the “Challenges” section unveils an imprecise view by the authors of the workflow in developing new biomarkers that includes an ordinate series of mandatory steps, before moving from benchtop to bedside. The authors should revise it, to help the reader in comprehension of the state of art of liquid biopsy.

Specific comments

Line 123: Probably, the authors intended that CTCs can be characterized by they morphology, phenotype and genotype..... Please, use simple and clear definitions.

Line 164: Cite here, please, the pivotal studies of Cohen (ref 83) and de Bono (ref 84) that determined the approval by FDA for using CellSearch platform in colorectal and prostate metastatic cancer, respectively.

Lines 165-170: Why do the authors discuss here the CellSearch data of the study of Dizdar et al.? Why they did not comment Gilupi data? This comparative study merits to be commented in another paragraph of the review.

Line 255: The authors cite the potential use of liquid biopsy in screening activity, but the cited paper (ref. 142) has been performed in COPD, in order to prevent lung cancer in high-risk patients… the authors should adequately comment this point.

Line 308: Here and throughout the manuscript, the authors should indicate the sample size of the cited studies, allowing the reader to evaluate soundness of the findings.

Lines 336-340: Why more studies should be done if the prognostic value has been yet provided? The authors should give other explanations of the rare use in clinical of the CellSearch assay.

Line 354: The Dizdar study should be cited here (see comment in lines 165-170). Moreover, the authors lack to cite other procedures that allow at increasing the analysis volume….

Line 362: The sentence is misleading. The phases of biomarkers development include a series of subsequent demonstrations, namely analytical validation, clinical validation and clinical utility…. Accuracy and reproducibility are part of the analytical validation.

Line 378: Prediction of OS and PFS is matter of clinical validity…

Line 401: This reviewer strongly disagree with summarizing here diagnostic and prognostic capability of liquid biopsy (see also general comment, please), since it might induce misleading concepts in the reader, in particular if she/he is not involved in this research area.

Author Response

We would like to express our gratitude to all the reviewers for their detailed reports. We found your comments and suggestions valuable and overall, they helped to increase the factual and stylistic precision of the text. Particularly, we have focused on improving the general understanding and logical coherence of the underlying theme of the review. Please see below for our direct replies.

Reviewer #1

General comment

The review of Kolenčík and colleagues aimed at revising the literature on CTCs in colon cancer. In the liquid biopsy research area, colon cancer is effectively a less frequented topic; hence, the review could be an opportunity to do the point on state of art of liquid biopsy in this malignancy. However, in the opinion of this reviewer, to address this purpose some gaps need to be filled before publication.

Speaking about the different technologies, the authors lack to summarize the main strategies of liquid biopsy procedures, and their pros and cons. Moreover, sometime ignore some fundamental findings.

RESPONSE: We deeply appreciate your comment regarding Liquid Biopsy Platforms section. We agree that providing more information would increase the impact of the review and we have decided to change section 3.1. Detection of Circulating Tumor Cells to capture fundamental findings more completely. Nevertheless, our main objective of the manuscript was to provide a comprehensive summarization of clinical applications and their challenges as opposed to a comprehensive summarization of technologies for liquid biopsy analysis.

For example, the authors choose to present a Table with the list of the technologies for CTC detection; furthermore, they discuss more in depth only three techniques. It might be a good way, to give a simple and precise picture of the state of art. Unfortunately, the reader is not warned that the list is largely incomplete (e.g. authors forget to cite DepArray), since liquid biopsy is a hot topic, and monthly several new techniques are published.

RESPONSE: Thank you for this review. We do agree that liquid biopsy is a very popular topic, leading to a very frequent development of new technologies. We have updated Table 1 in the text with additional platforms to highlight more technologies.

Moreover, the authors do not explain why they choose to discuss CellSearch, ISET and Epic Science technologies. Overall, the section about CTC detection techniques results poor. 

RESPONSE: Thank you for this critique. As we have stated above, our main objective with this review was to summarize the clinical applications and challenges of liquid biopsy technologies. As such, we focus on these three technologies to highlight some of the more common approaches. We have adjusted the section 3.1. Detection of Circulating Tumor Cells to add more clarity for the reader.

Furthermore, despite the authors underline the value of liquid biopsy to investigate temporal and spatial tumor heterogeneity, they lack to discuss some sound papers, e.g. those of Fabbri and Bardelli, in colon cancer.

RESPONSE: Thank you. We have discussed sound papers in section 4. Clinical applications. We believe this information will point out the importance of intratumor heterogeneity and the value of liquid biopsy.

Second, in the opinion of this reviewer the clinical section of the review is not well organized; moreover, there are semantic imprecisions in description of clinical activity, hence inducing misleading conclusions in the reader. For example, the first paragraph is entitled “Diagnostics”; in this paragraph, there is an overlap between early diagnosis and early prediction of treatment effectiveness, that are different uses of biomarker. By the way, there are no prospective studies about the diagnostic use of liquid biopsy in colon cancer, whilst the use of liquid biopsy as predictor of treatment effectiveness has been partially entered in colon cancer management.

RESPONSE: We truly appreciate this comment. We agree that section 4.1. Diagnostics does not solely describe diagnostic use of the liquid biopsy. To maintain clarity and consistency, we have moved a portion of the paragraph discussing variation of KRAS and NRAS mutations into section 4.2. Treatment Selection.

Finally, the “Challenges” section unveils an imprecise view by the authors of the workflow in developing new biomarkers that includes an ordinate series of mandatory steps, before moving from benchtop to bedside. The authors should revise it, to help the reader in comprehension of the state of art of liquid biopsy.

RESPONSE: We appreciate this comment as our objective was to provide not just clinical applications but also report on limitations in an intelligible way. As a result, we have rearranged section 5. Challenges to make these points more clearly. 

Specific comments

Line 123: Probably, the authors intended that CTCs can be characterized by they morphology, phenotype and genotype..... Please, use simple and clear definitions.

RESPONSE: We appreciate this critique. We have phrased this sentence with more definite terminology.

Line 164: Cite here, please, the pivotal studies of Cohen (ref 83) and de Bono (ref 84) that determined the approval by FDA for using CellSearch platform in colorectal and prostate metastatic cancer, respectively.

RESPONSE:  Thank you. These references are now cited.

Lines 165-170: Why do the authors discuss here the CellSearch data of the study of Dizdar et al.? Why they did not comment Gilupi data? This comparative study merits to be commented in another paragraph of the review.

RESPONSE:  Thank you for this comment. We agree with the suggestion and have relocated this discussion about the study to section 4.3. Prognostics, in which it is better suited. Additionally, we have expanded the discussion to include that of the Gilupi data.

Line 255: The authors cite the potential use of liquid biopsy in screening activity, but the cited paper (ref. 142) has been performed in COPD, in order to prevent lung cancer in high-risk patients… the authors should adequately comment this point.

RESPONSE: Thank you for this comment. We have replaced the reference with more adequate ones.

Line 308: Here and throughout the manuscript, the authors should indicate the sample size of the cited studies, allowing the reader to evaluate soundness of the findings.

RESPONSE: We have reviewed all cited studies and added sample size information where appropriate.

Lines 336-340: Why more studies should be done if the prognostic value has been yet provided? The authors should give other explanations of the rare use in clinical of the CellSearch assay.

RESPONSE: We appreciate the critique and we acknowledge the current phrasing of these two statements result in a contradiction. We have rephrased the last sentence of the paragraph to clarify that we observe clinical validity but limited clinical utility and therefore other studies of clinical utility is warranted. In this setting, we are using the terms “clinical validity” and “clinical utility” as described by Scher et al. in Validation and clinical utility of prostate cancer biomarkers.

Line 354: The Dizdar study should be cited here (see comment in lines 165-170). Moreover, the authors lack to cite other procedures that allow at increasing the analysis volume….

RESPONSE: Thank you. We agree that Dizdar study better fits in section 5. Challenges. We have cited the study in this paragraph and further discuss it.

Line 362: The sentence is misleading. The phases of biomarkers development include a series of subsequent demonstrations, namely analytical validation, clinical validation and clinical utility…. Accuracy and reproducibility are part of the analytical validation.

RESPONSE: Thank you for this comment. We acknowledge the sentence is misleading and we have changed the phrase “clinical utility” for “analytic validity” to address the confusion.

Line 378: Prediction of OS and PFS is matter of clinical validity…

RESPONSE: As with our previous response, we agree the statement was confusing, and we have changed to “clinical validity and utility”.

Line 401: This reviewer strongly disagree with summarizing here diagnostic and prognostic capability of liquid biopsy (see also general comment, please), since it might induce misleading concepts in the reader, in particular if she/he is not involved in this research area.

RESPONSE: Thank you for this comment. We agree that the specific sentence in Line 401 poses a question which has not yet been discussed and does not fit with the rest of the section. We have changed the sentence to create a more coherent paragraph that fits with the overall message of the section.

Reviewer 3 Report

This review by Kolencik and colleagues summarizes the major characteristics, analysis methods and applications of liquid biopsies (CTCs and ctDNA).

Although the information reported are mostly correct (see details below), they are quite dated and the work doesn't add anything to the many other reviews already published. In particular, there's no critical discussion of the most novel uses of liquid biopsy in the context of minimal residual disease, or the definition of lesion-specific responses to a certain therapy.

I would suggest the authors to reorganize their work highlighting these findings.

Also, these provided information should be revised:

1) active release of circulating free DNA can occur not only from lymphocites but also from circulating tumor cells

2) DNA fragments produced during apoptosis processes are around 150 bp

3) technologies aimed at detecting single tumor alterations are not only dedicated to point mutations, as small indels and specific CNA can be identified by ddPCR

4) the MAF is the result of the number of mutant alleles on the TOTAL of target alleles in the reaction (mutant plus WT), not on WT only

5) Amplicon-based sequencing is (too) briefly discussed, while there's no mention to targeted panels with hybrid capture methods, even with the use of duplex sequencing and UMI molecular barcodes. These technologies have recently improved ctDNA detection sensitivity and should be greatly discussed

6) contrary to what is stated, there is a FDA approved ctDNA test for CRC (late 2019, Natera)

7) when discussing the current challenges in standardization of procedures, there's no mention to sample collection protocols

Overall, I believe the author should consider working on rewriting the review before it's considered for publication.

Author Response

We would like to express our gratitude to all the reviewers for their detailed reports. We found your comments and suggestions valuable and overall, they helped to increase the factual and stylistic precision of the text. Particularly, we have focused on improving the general understanding and logical coherence of the underlying theme of the review. Please see below for our direct replies.

Reviewer #2

Important:

This review by Kolencik and colleagues summarizes the major characteristics, analysis methods and applications of liquid biopsies (CTCs and ctDNA).

Although the information reported are mostly correct (see details below), they are quite dated and the work doesn't add anything to the many other reviews already published. In particular, there's no critical discussion of the most novel uses of liquid biopsy in the context of minimal residual disease, or the definition of lesion-specific responses to a certain therapy.

RESPONSE:  We deeply appreciate your comment and we have added information about lesion-specific response to targeted therapy and the use of liquid biopsy in context of minimal residual disease.

I would suggest the authors to reorganize their work highlighting these findings.

Also, these provided information should be revised:

1) active release of circulating free DNA can occur not only from lymphocites but also from circulating tumor cells

RESPONSE: We have corrected this inaccuracy in section 2.2. Cell-free DNA.

2) DNA fragments produced during apoptosis processes are around 150 bp

RESPONSE: We have corrected this inaccuracy in section 2.2. Cell-free DNA.

3) technologies aimed at detecting single tumor alterations are not only dedicated to point mutations, as small indels and specific CNA can be identified by ddPCR

RESPONSE: Thank you for this comment. We have added a small section describing the genomics technologies in more detail.

4) the MAF is the result of the number of mutant alleles on the TOTAL of target alleles in the reaction (mutant plus WT), not on WT only

RESPONSE: Thank you for this comment. The original sentence is misleading and was changed so that it would comply with correct definition of MAF.

5) Amplicon-based sequencing is (too) briefly discussed, while there's no mention to targeted panels with hybrid capture methods, even with the use of duplex sequencing and UMI molecular barcodes. These technologies have recently improved ctDNA detection sensitivity and should be greatly discussed

RESPONSE: Proportionate information about duplex sequencing and UMI molecular barcodes has been added to the section 3.2. cfDNA analysis. Furthermore, we have discussed other types of method for analyzing cfDNA.

6) contrary to what is stated, there is a FDA approved ctDNA test for CRC (late 2019, Natera)

RESPONSE: We appreciate this comment, which led us to finding a contradiction in our text. In response, we have removed the contradictory sentence in section 4.3 Prognostics, and we have added a reference for an FDA-approved assay in CRC. We have also included Signatera™ in section 3.2. as another example of FDA-approved assay for CRC.

7) when discussing the current challenges in standardization of procedures, there's no mention to sample collection protocols

RESPONSE: Thank you. We have added information about the necessity of sample collection protocols.

Overall, I believe the author should consider working on rewriting the review before it's considered for publication.

Round 2

Reviewer 3 Report

I thank the authors for addressing my concerns and modifying their work as per my suggestions.